**Data Availability Statement:** All data are included within the manuscript.

**Funding:** The research presented here was supported by funding from the Mayo Clinic

# Identification of naturally processed Zika virus peptides by mass spectrometry and validation of memory T cell recall responses in Zika convalescent subjects

Stephen N. Crooke, Inna G. Ovsyannikova, Richard B. Kennedy, Gregory A. Poland*

Mayo Clinic Vaccine Research Group, Mayo Clinic, Rochester, Minnesota, United States of America

* poland.gregory@mayo.edu

## Abstract

Once an obscure pathogen, Zika virus (ZIKV) has emerged as a significant global public health concern. Several studies have linked ZIKV infection in pregnant women with the development of microcephaly and other neurological abnormalities, emphasizing the need for a safe and effective vaccine to combat the spread of this disease. Preclinical studies and vaccine development efforts have largely focused on the role of humoral immunity in disease protection. Consequently, relatively little is known in regard to cellular immunity against ZIKV, although an effective vaccine will likely need to engage both the humoral and cellular arms of the immune system. To that end, we utilized two-dimensional liquid chromatography coupled with tandem mass spectrometry to identify 90 ZIKV peptides that were naturally processed and presented on HLA class I and II molecules (HLA-A*02:01/HLA-DRB1*04:01) of an immortalized B cell line infected with ZIKV (strain PRVABC59). Sequence identity clustering was used to filter the number of candidate peptides prior to evaluating memory T cell recall responses in ZIKV convalescent subjects. Peptides that individually elicited broad (4 of 7 subjects) and narrow (1 of 7 subjects) T cell responses were further analyzed using a suite of predictive algorithms and *in silico* modeling to evaluate HLA binding and peptide structural properties. A subset of nine broadly reactive peptides was predicted to provide robust global population coverage (97.47% class I; 70.74% class II) and to possess stable structural properties amenable for vaccine formulation, highlighting the potential clinical benefit for including ZIKV T cell epitopes in experimental vaccine formulations.

## Importance

Zika virus (ZIKV) infection in pregnant women has been implicated as a causative agent of microcephaly, fetal birth defects, and other neurological abnormalities. ZIKV infection has been associated with increased cases of Guillain-Barré Syndrome (GBS), which is characterized by muscle weakness, motor dysfunction and, in some cases, paralysis. There is currently no

Department of General Internal Medicine and the Maurice R. Hilleman Early-Stage Career Investigator Award to SNC (National Foundation for Infectious Diseases and Merck & Co. Inc.). The funders had no role in study design, data collection and analysis, decision to publish, or preparation of the manuscript.

**Competing interests:** Dr. Poland is a paid scientific advisor for Johnson & Johnson/Janssen Global Services LLC. Dr. Poland is the chair of a Safety Evaluation Committee for novel investigational vaccine trials being conducted by Merck Research Laboratories. Dr. Poland offers consultative advice on vaccine development to Merck & Co., Medicago, GlaxoSmithKline, Sanofi Pasteur, Emergent Biosolutions, Dynavax, Genentech, Eli Lilly and Company, Kentucky Bioprocessing, Bavarian Nordic, AstraZeneca, Exelixis, Regeneron, Janssen, Vyriad, Moderna, and Genevant Sciences, Inc. Drs. Poland and Ovsyannikova hold patents related to vaccinia and measles peptide vaccines. Dr. Kennedy holds a patent related to vaccinia peptide vaccines. Drs. Poland, Kennedy, and Ovsyannikova have received grant funding from ICW Ventures for preclinical studies on a peptide-based COVID-19 vaccine. Dr. Kennedy has received funding from Merck Research Laboratories to study waning immunity to mumps vaccine. Dr. Crooke received funding from the National Foundation for Infectious Diseases and Merck & Co. Inc. to support this research. This does not alter our adherence to PLOS ONE policies on sharing data and materials. These activities have been reviewed by the Mayo Clinic Conflict of Interest Review Board and are conducted in compliance with Mayo Clinic Conflict of Interest policies. This research has been reviewed by the Mayo Clinic Conflict of Interest Review Board and was conducted in compliance with Mayo Clinic Conflict of Interest policies. All other authors declare no competing financial interests.

licensed vaccine available to provide protection against this disease. In this study, we utilized a tandem mass-spectrometry approach to identify viral peptides that were processed during viral infection, and we validated the immunogenicity of a subset of these peptides by demonstrating memory recall immune responses in PBMCs from ZIKV convalescent subjects. Moreover, computational analyses predicted that these peptides would provide broad global-population coverage, which suggests that they may be useful components of a peptide-based ZIKV vaccine. Our mass-spectrometry-based approach is a powerful tool for 1) identifying peptide epitopes that are naturally recognized by the immune system and 2) informing the design of experimental vaccines against emerging pathogens.

## Introduction

Originally identified in Uganda in the Zika forest in 1947 [1], Zika virus (ZIKV) is a 10-kilo-base single-stranded, positive-sense RNA virus of the flavivirus family that remained a relatively obscure and unimportant pathogen until large outbreaks in French Polynesia (2013–14) and Brazil (2015–16) coincided with increased diagnoses of severe neurological abnormalities —particularly among newborn children [2–6]. Strong evidence has since confirmed the association between ZIKV infection in pregnant women and the development of fetal malformations (e.g., microcephaly, congenital contractures, hypertonia, macular and retinal damage) [7, 8]; in severe cases, *in utero* ZIKV infection can result in premature fetal death. In contrast, ZIKV infections in healthy adults are relatively mild and often asymptomatic with self-limited illness lasting two to seven days [9]. Nevertheless, the incidence of GBS among older adults rose sharply in conjunction with ZIKV outbreaks in Brazil and French Polynesia, [4, 10] indicating these individuals are also at risk for ZIKV-associated neurological complications. Furthermore, several reports have documented the persistent detection of ZIKV RNA in semen several months after infection and the sustained potential for sexual transmission [11, 12], further emphasizing the need for a vaccine even in non-outbreak settings.

ZIKV is commonly transmitted by *Aedes aegytpi* and *Aedes albopictus* mosquito vectors during their feeding cycles [13]; however, incidences of sexual and perinatal transmission from infected individuals have also been reported [14, 15]. Due to the geographic distribution of *Aedes* spp. mosquitoes, ZIKV outbreaks have been primarily limited to tropical and subtropical climates, although isolated cases of autochthonous transmission have been reported in the southern United States and Europe [16, 17]. Recent estimates indicate that 60% of the US population currently resides within the geographic ranges of *Aedes* spp. mosquitoes, and increases in global temperatures as a result of climate change threaten to further promote the spread of *Aedes* mosquitoes—and the transmission of ZIKV—into temperate regions [13, 18]. The spread of ZIKV among immunologically naïve populations may have devastating effects, emphasizing the need for a safe and effective vaccine.

As of March 2016, 18 ZIKV vaccine candidates were reportedly in various stages of development, although none have since progressed beyond Phase II clinical trials as government funding has been redirected in the wake of subsiding ZIKV outbreaks [19, 20]. Plasmid DNA-based vaccines encoding the ZIKV pre-membrane (M) and envelope (E) proteins advanced the farthest in clinical testing, demonstrating adequate safety profiles and eliciting both humoral and cellular immune responses to varying degrees [21].

Mechanistic studies in mice and non-human primates have shown humoral immunity to be sufficient for protection against ZIKV infection [22–24], but the role of cellular immunity remains unclear. Cellular immune responses contribute to viral clearance and protection against other flaviviruses [25, 26], suggesting that ZIKV-specific T cell responses may also be important. A study by Elong Ngono et al. demonstrated that adoptive transfer of ZIKV-

specific CD8[+] T cells reduced viral burden in mice [27]. Similarly, Hassert and colleagues reported that polyfunctional CD4[+] T cell responses were critical for protection against disease in a mouse model of ZIKV infection [28]. While these studies suggest that cellular immunity plays a critical role in mediating protection against ZIKV, reports on ZIKV-specific T cell responses in humans are lacking, and no studies to-date have analyzed epitopes recognized by human T cells during ZIKV infection. It is imperative that we develop a clear understanding of the ZIKV immunopeptidome in order to engineer vaccines capable of eliciting robust humoral *and* cellular immune responses.

Here, we employed nanoscale liquid chromatography coupled with tandem mass spectrometry (nLC-MS/MS) to identify 90 peptides from structural and non-structural ZIKV proteins that were naturally processed and presented on human leukocyte antigen (HLA) class I and class II molecules during *in vitro* infection. We subsequently evaluated memory T cell recall responses from ZIKV convalescent subjects against a refined subset (n = 34) of these peptides to confirm their immunological relevance and used bioinformatics to characterize a selection of nine peptides as potential ZIKV vaccine components.

## Methods

The methods described herein are the same or similar to those in our previous publications [29–31].

### Convalescent subjects

Peripheral blood mononuclear cells (PBMCs) from seven healthy donors (four male, three female) with prior documented ZIKV infection were provided courtesy of the National Institutes of Health Vaccine Research Center. Subjects (26–39 years of age) were recruited between February and August 2016 and participated in a blood draw ~ 21–138 days post-infection. Samples were deidentified and are herein denoted by an arbitrary numerical study identifier: 591, 596, 602, 625, 626, 627, and 629. Subject 591 provided samples at two timepoints post-infection: ~ 21 days and ~ 138 days. These samples are subsequently denoted 591–1 and 591–3, respectively. All study participants provided written informed consent, and all recruitment procedures were approved by the National Institute of Allergy and Infectious Diseases Institutional Review Board.

### Cell culture and ZIKV infection

ZIKV (Puerto Rico strain PRVABC59; ATCC #VR-1843) was propagated in C6/36 cells (ATCC #CRL-1660) in minimum essential medium (MEM) supplemented with 2% fetal calf serum (FCS; Life Technologies; Gaithersburg, MD). An unverified immortalized B cell line (Priess; Millipore Sigma #86052111) homozygous for HLA-A*02:01 and HLA-DRB1*04:01 was used for this study. The complete HLA profile for Priess cells (as reported by the European Collection of Authenticated Cell Cultures) is HLA-A*02:01, HLA-B*15, HLA-DRB1*04:01, HLA-DRB4*01:01, HLA-DQA1*03, HLA-DQB1*03, HLA-DPA1*01:03, HLA-DPB1*03:01, HLA-DPB1*04:01. Priess cells (~ 4 x 10$^8$) were infected with ZIKV at a multiplicity of infection (MOI) of 0.5 for 2 hours and subsequently maintained for 5 days in MEM supplemented with 2% FCS. Negative control cell cultures consisted of Priess cells mock-infected with virus-free PBS.

### Peptide isolation and fractionation

Peptides presented on class I and class II HLA molecules of ZIKV-infected Priess cells were isolated as previously described [29–33]. Briefly, cells were washed twice with PBS and treated

for 3 min with acidic citrate-phosphate buffer (0.066 M $Na_2HPO_4$, 0.13 M citric acid, 290 mOsmol/kg $H_2O$, pH 3.0) to denature surface-expressed HLA-peptide complexes [34]. Peptides were purified from HLA molecules by centrifugal filtration through a prewashed Centricon-10 kDa molecular weight cut-off filter (Millipore, Bedford, MA). Salts were removed from the peptide mixture by washing with 2% acetonitrile in 0.1 M acetic acid using a reversed phase 1 mm x 8 mm peptide trap (Michrom BioResources; Auburn, CA). Peptides were eluted with 60% acetonitrile in 0.1 M acetic acid, vacuum-concentrated, and reconstituted in 5 mM $KH_2PO_4$ (pH 3.0). Strong cation exchange (SCX) fractionation was performed on the desalted peptides using a gradient of 0–0.4 M KCl in 5 mM $KH_2PO_4$/20% acetonitrile (pH 3.0). Peptides were loaded onto a polysulfoethyl aspartamide column (Michrom BioResources) and separated on a 0–0.2 M KCl linear gradient over 20 min followed by a 0.2–0.4 M KCl linear gradient over 10 min. Fractions were collected at 2-min intervals and stored at –80˚C until analysis.

## Peptide analyses by nLC-MS/MS

SCX fractions were thawed, vacuum-concentrated, and reconstituted with 5% acetonitrile in 5 mM $KH_2PO_4$ (40 μL; pH 3.0). All measurements were performed on a linear ion trap-Fourier transform hybrid mass spectrometer (LTQ-Orbitrap, Thermo Fisher Scientific, Waltham, MA) interfaced with a 15 cm x 75 μm Magic $C_{18AQ}$ column (Michrom BioResources) on a nano-scale liquid chromatograph and autosampler (Eksigent NanoLC 1D, Dublin, CA). Mobile phase A was comprised of water/acetonitrile/formic acid (98/2/0.2% v/v), and mobile phase B was comprised of acetonitrile/water/formic acid (90/10/0.2% v/v). Samples (5–20 μL) were first loaded onto a Magic $C_8$ pre-column (Michrom BioResources) with 0.05% trifluoroacetic acid/0.15% formic acid in water at 15 μL/min. Samples were run at 0.4 μL/min for 90 min employing a gradient of 2–40% mobile phase B over 60 min, with ramping to 90% mobile phase B at 85 min.

SCX fractions were subsequently analyzed by nLC-MS/MS using data-dependent acquisition parameters, including an Orbitrap survey scan with 60,000 resolving power, a target population of 1 x $10^6$ ions, and a maximum fill time of 300 ms. Fourier transform was used to select the most abundant ion species for MS/MS analysis. LTQ MS/MS spectra were acquired with 2.5 mass unit isolation width, a target ion population of 1 x $10^4$ ions, one microscan, 100 ms maximum fill time, 35% normalized collision energy, activation Q of 0.25, and 30 ms activation time. Ions selected for MS/MS were excluded for 45 sec, with an exclusion window of 1 *m/z* below and 1.6 *m/z* above the exclusion mass. Singly charged species were identified between 700–1500 *m/z* and doubly/triply-charged species were identified between 340–1200 *m/z*, consistent with the average molecular weight of most HLA-presented peptides.

## MS/MS data analyses

All queries were run against a subset of the SwissProt database (July 2017) containing human, bovine, and ZIKV proteins from two UniProt accessions. The first is SwissProt (curated) accession Q32ZE1, POLG ZIKV, which is the linear form of the African-origin virus (3423 amino acids) and is annotated with the cleavage points for the various protein products. This entry has 96.4% homology with ZIKV nucleotide accession KX377337.1. Using information on the cleavage sites, we also created database entries for each of the protein products. The second is TREMBL (non-curated) accession A0A192GPS0 ZIKV, which is the Puerto Rico strain and has 100% homology with the nucleotide accession KX377337.1 but does not contain information for the cleavage sites of the individual protein products. Bovine proteins were included in the database search because cell culture media contained fetal calf serum. Randomized

protein sequences were included in the database as decoys to estimate the false positive rate. All search queries were run with a mass tolerance of 7 parts per million (ppm), 0.6 fragment ion mass unit tolerance, and ignoring protease specificity. Methionine oxidation was considered as a potential modification. Results from all analyses of SCX fractions were curated and exported into an Excel spreadsheet. The final list of identified peptides was comprised of all peptides identified between two biological replicate experiments.

## IFN-γ T cell ELISpot

ZIKV peptides identified by nLC-MS/MS were individually synthesized in large batches (5 mg) by GenScript (Piscataway, NJ) for functional testing. Peptide sequences were filtered using the IEDB Clustering Tool, with the clustering threshold set at 80% sequence identity. The filtered peptides were arbitrarily assigned a numerical identifier (1–34) and randomly sorted into six pools of eight to nine peptides, with each pool containing three unique peptides and six peptides duplicated evenly across two other pools. Polyvinylidene fluoride-backed 96-well microtiter plates were coated with anti-human IFN-γ overnight at 4˚C. Plates were washed thrice with PBS-Tween 20 (0.05%) and blocked for 2 hrs with DMEM supplemented with 10% FCS. PBMCs were seeded (2 x $10^5$ cells/well) and treated with one of the following conditions: DMEM culture media (unstimulated); 20 μg pooled ZIKV peptides; live ZIKV (MOI = 1); or 20 μg pooled human actin peptides (JPT Peptide Technologies Inc.; Acton, MA) as a negative control. Cells were incubated for 18 hours and T cell responses quantified using human IFN-γ ELISpot kits (BD Biosciences; San Jose, CA) according to the manufacturer's protocol. For subsequent screening of individual peptides, the same procedure was followed using 10 μg ZIKV peptides for stimulation. Samples stimulated with ZIKV peptides or live virus were tested in triplicate; unstimulated samples and negative controls were tested in quadruplicate. Plate images were captured and analyzed using an ImmunoSpot S6 Core Analyzer (Cellular Technology Limited; Cleveland, OH), with responses quantified as spot-forming units (SFUs) per 2 x $10^5$ cells. The limit of detection was set at 2 standard deviations from the mean SFU count for unstimulated PBMCs. All ELISpot responses were evaluated by paired t-test.

## HLA binding predictions

The amino acid sequences of seven proteins (E, capsid [C], non-structural protein 1 [NS1], NS2A, NS3, NS4B, NS5) from the ZIKV PRVABC59 strain (GenBank accession: AWH65849) were downloaded from the Virus Pathogen Resource database (https://www.viprbrc.org/brc/viprStrainDetails.spg?strainName=PRVABC59&decorator=flavi) and used as inputs for predictive calculations of HLA class I and class II peptide binding. Sequences for the M, NS2B, and NS4A proteins were excluded from analysis as no peptides identified by nLC-MS/MS mapped to these proteins.

For class I peptide predictions, the NetMHCpan 4.0 (Immune Epitope Database) algorithm was used to predict binding affinities and score peptide binding metrics across the most frequently occurring HLA-A and HLA-B alleles [35, 36]. Peptide lengths were restricted to 8–14 amino acids. The complete list of predicted peptides was filtered for consensus sequences matching (or nested within) select peptide sequences identified from nLC-MS/MS and ELISpot analyses. When an exact match or complete nested sequence was not identified, peptides with partial sequence homology to the predicted consensus sequence were selected. The threshold for binding was set at 5%.

For class II peptide predictions, complete protein sequences were analyzed using either NetMHCIIpan 3.2 or a Consensus method (IEDB) which evaluates peptide binding using a

combination of artificial neural networks (NN-align 2.3), stabilized matrices (SMM-align), combinatorial libraries (CombLib) and Sturniolo [37–41]. Peptide binding predictions were performed against a reference panel of 27 HLA class II alleles [42]. Peptide sequences shorter than 12 amino acids were excluded from analysis. The complete list of predicted peptides was filtered for consensus sequences matching (or nested within) select peptide sequences identified from nLC-MS/MS and ELISpot analyses. When an exact match or complete nested sequence was not identified, peptides with partial sequence homology to the predicted consensus sequence were selected. The threshold for binding was set at 10%.

## Peptide structural modeling

The predicted structures of select peptides were modeled using PEP-FOLD 3.5, an online server that predicts peptide structure based on the properties of each amino acid in the sequence [43–47]. Properties of the selected peptides were determined *in silico* using the Protparam tool hosted on the ExPASy server [48].

## Population coverage analysis

The Population Coverage tool available through IEDB was used to estimate the population coverage of select peptides across 16 distinct geographic regions as defined by allele frequencies in the Allele Frequency database [49]. The alleles for each peptide epitope were selected based on HLA binding prediction data, and calculations were limited to the top 2% of predicted alleles (corresponding to ~ 5 alleles per peptide).

# Results

## Identification of ZIKV peptides by nLC-MS/MS

The workflow used to identify HLA-presented viral peptides following ZIKV infection is outlined in Fig 1. Six SCX fractions were analyzed by nLC-MS/MS, resulting in 2,305 MS/MS spectra that were subsequently cross-referenced against human, bovine, and ZIKV protein sequences listed in the SwissProt database (July 2017). From this analysis at the 0.1% local false discovery rate, we identified 90 peptides derived from viral proteins: 59 from NS1; two from NS2A; seven from NS3; four from NS4B; eight from NS5; eight from the capsid; and two from the E protein. The distribution of peptide lengths is presented in Fig 2A. Peptides were putatively classified based on the canonical sequence lengths for HLA class I (7–15 amino acids) and class II ($\geq$ 13 amino acids) peptides. The majority of class I peptides were 9–12mers, whereas class II peptides exhibited a bimodal length distribution between 13–17 amino acids and 20–24 amino acids. In order to conserve biological samples at the outset of peptide screening, peptides with high degrees of sequence similarity (irrespective of class I or class II classification) were grouped into clusters (Fig 2B). We identified 27 unique clusters (including singletons) and selected 34 unique peptides (Table 1) for further evaluation in biologic assays. Peptides were selected on the basis of unique sequence identity, with at least one peptide selected from each cluster. Multiple peptides were selected from larger clusters based on homology with the consensus sequence of the cluster.

## Memory T cell recall responses in convalescent subjects

Peptides were randomly sorted into pools as indicated in Table 1 and used to evaluate memory T cell responses in PBMCs from convalescent subjects (Fig 3A and 3B). Peptides in pools 5 and 6 stimulated strong average IFN-γ responses (12.9 and 9.8 SFUs/2 x 10$^5$ cells, respectively) in six of the seven subjects tested, while peptides in pool 2 stimulated a strong

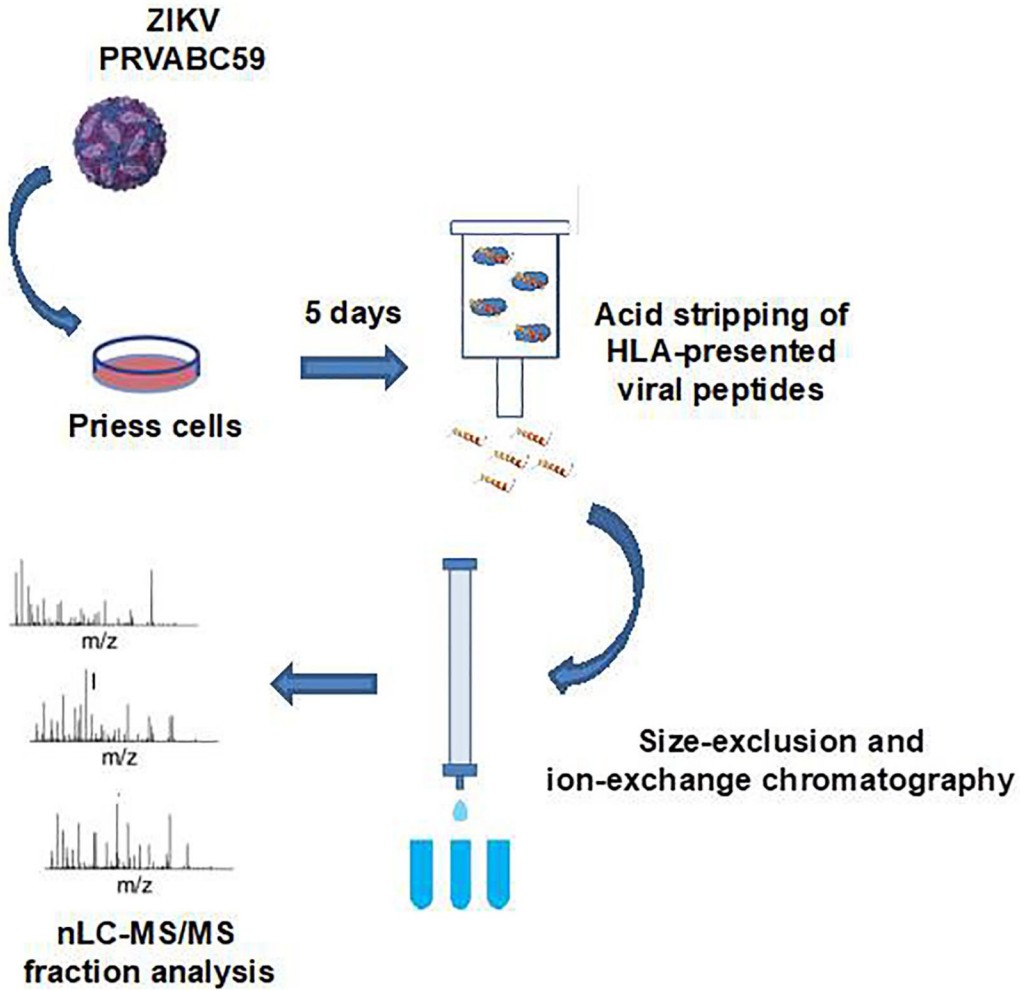

**Fig 1. Peptide identification workflow.** Priess cells were infected with ZIKV (PRVABC59 strain) and viral peptides presented on HLA molecules were isolated using an acidic wash followed by size-exclusion and strong cation exchange chromatography. Peptide fractions were analyzed by nanoscale liquid chromatography-coupled tandem mass spectrometry.

response (10.7 SFUs/2 x $10^5$ cells) in five subjects. Weaker IFN-γ responses were observed on average (3.7, 4.7, and 3.8 SFUs/2 x $10^5$ cells) for peptides in pools 1, 3, and 4, respectively. Responses were highly variable between individual subjects (Fig 3C). Due to this variability, none of the ELISpot responses to the peptide pools met statistical significance compared to background, although it was evident that several individual responses were significant. Average ELISpot responses to pools 5 and 6 were statistically comparable ($p = 0.12$, $p = 0.06$) to responses against ZIKV. Subject 602 exhibited robust T cell responses to all the peptide pools, whereas subject 627 only exhibited marginally detectable responses to pools 3 and 4 (~ 2 SFUs/2 x $10^5$ cells). Notably, all subjects exhibited strong IFN-γ responses to live ZIKV irrespective of their response to the pooled peptides. There was no significant correlation between the time since infection and the magnitude of the T cell response (S1 Fig), although T cell responses for subject 591 had significantly declined by the second blood draw (sample 591–3; Fig 3C).

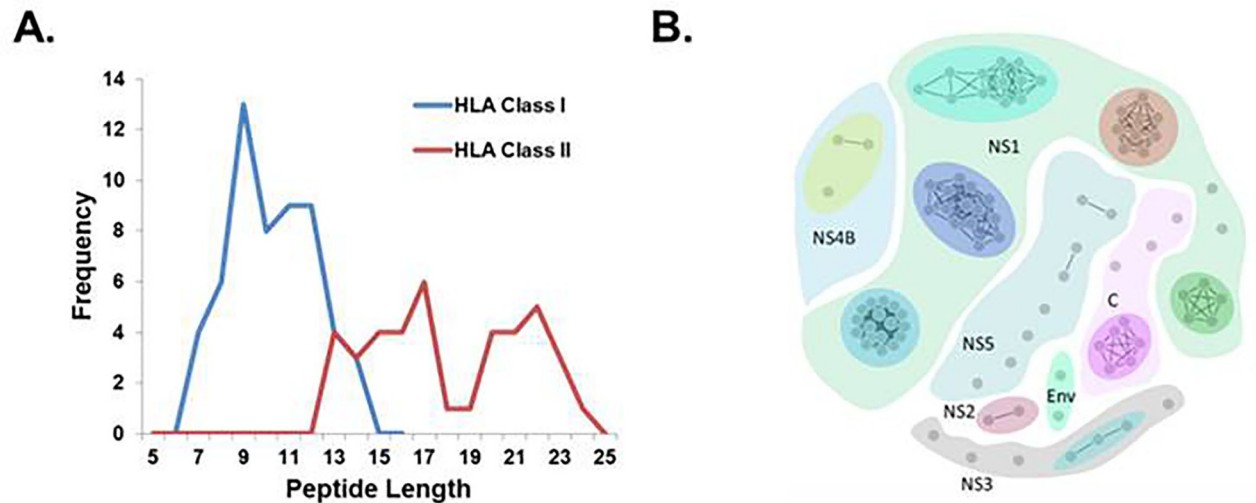

**Fig 2. Characterization of isolated ZIKV peptides.** (A) Distribution of peptide length for ZIKV peptides identified by mass spectrometry. Peptides were assigned as HLA class I or class II binders based on the canonical length of peptides that bind molecules of each respective HLA class. (B) Sequence homology clustering of ZIKV peptides. The clustering threshold for sequence identity was set at 80%. Each grey node represents a unique peptide, and arbitrarily colored divisions delineate the viral protein from which peptide clusters were derived.

In order to identify the specific ZIKV peptides stimulating memory T cell recall responses, we individually analyzed each peptide from the pools that had stimulated an IFN-γ response in a given subject (Fig 3D). Similar to our observations with the peptide pools, some subjects exhibited broad recognition of several peptide epitopes while others exhibited very limited recognition of only a few peptides. We identified nine peptides (three from NS1, two from C, two from NS3, and two from NS5) that individually stimulated a T cell recall response in four of seven subjects (Fig 3E). These "broadly reactive" peptides were selected as leading candidates for detailed informatics analysis. Eight peptides (four NS1, two NS4B, two NS5) that individually stimulated a narrow recall response in only one of seven subjects were selected as comparators (Fig 3E). Subjects 602 and 625 were the strongest responders to peptides in the comparator group, with subject 625 responding to four of the eight peptides. Three subjects (626, 596, and 629) did not respond to any of the peptides in the comparator group. In contrast, all subjects exhibited recall responses to a minimum of three peptides in the candidate group, illustrating the breadth of coverage provided by the candidate peptide pool.

## Predictions of HLA binding and global population coverage

As our access to biospecimens from convalescent subjects was limited, we used predictive algorithms in order to assess the full breadth of HLA alleles that our candidate peptides would bind. The complete list of HLA class I and class II alleles identified for both the candidate peptides as well as the comparator peptides is presented in Table 2. The candidate peptides were predicted to bind more broadly across the predicted HLA-A and HLA-B alleles relative to the comparator peptides (Fig 4A). In particular, eight of the nine candidate peptides were predicted to bind HLA-A*02:03 molecules. Candidate peptides were also enriched for binding to HLA-B molecules compared to comparator peptides, with several alleles (B*24:02, B*35:01, B*40:01, B*44:03) predicted to bind peptides solely from the candidate pool. Binding to HLA class II alleles was predicted to be far more limited compared to class I alleles across both the candidate and comparator peptide pools, although candidate peptides were predicted to bind a broader array of class II alleles relative to the comparator peptides (Table 2, Fig 4B).

**Table 1. ZIKV peptides selected from sequence homology clustering for testing.**

| Numerical Identifier | Peptide Sequence | Viral Protein | Peptide Pools |
|---|---|---|---|
| 1 | GRGPQRLPVP | NS1 | 1, 6 |
| 2 | ALALAIIKY | NS5 | 1, 6 |
| 3 | YLDKQSDTQYV | E | 1, 6 |
| 4 | RQDQRGSGQVVTY | NS5 | 1 |
| 5 | YQNKVVKVL | NS5 | 1 |
| 6 | TVTRNAGLVKRR | NS4B | 1 |
| 7 | DPAVIGTAVKGREAAH | NS1 | 1, 2 |
| 8 | AVQHAVTTSY | NS4B | 1, 2 |
| 9 | YLIPGLQAA | NS4B | 1, 2 |
| 10 | RLPAGLLLGHGPIRMVL | C | 2 |
| 11 | LIIPKSLAGPLSHHNTREG | NS1 | 2 |
| 12 | YLQDGLIASL | NS3 | 2 |
| 13 | LTVVVGSVKNPmGRGPQRLPVPVN | NS1 | 2, 3 |
| 14 | DPAVIGTAVKGKEAVHSDLG | NS1 | 2, 3 |
| 15 | IIPKSLAGPLSHHNTREGYRTQ | NS1 | 2, 3 |
| 16 | RGPQRLPVPVN | NS1 | 3 |
| 17 | LVEDHGFGVFHTSVW | NS1 | 3 |
| 18 | ALWDVPAPKEV | NS3 | 3 |
| 19 | ANPVITESTENSK | E | 3, 4 |
| 20 | KVRPALLVSF | NS2 | 3, 4 |
| 21 | RMLLDNIYL | NS3 | 3, 4 |
| 22 | LRFTAIKPSLGLINR | C | 4 |
| 23 | FKVRPALL | NS2 | 4 |
| 24 | TVVVGSVKNPMWRGPQRLPVPVN | NS1 | 4 |
| 25 | IMWRSVEGELNA | NS1 | 4, 5 |
| 26 | LAVPPGERARNIQTLPGIFK | NS3 | 4, 5 |
| 27 | FLRFTAIKPSLG | C | 5 |
| 28 | QVASAGITY | NS3 | 5 |
| 29 | SLINGVVRL | NS5 | 5 |
| 30 | GGLKRLPAGLLLGHGPI | C | 5, 6 |
| 31 | VVDGDTLK | NS1 | 5, 6 |
| 32 | DGIEESDLIIPKSLAGP | NS1 | 5, 6 |
| 33 | TMMETLERL | NS5 | 6 |
| 34 | RIIGDEEKY | NS5 | 6 |

Using the predicted binding alleles from Table 2, the global population coverage for the candidate and comparator peptides pools was estimated across 16 distinct geographic regions defined by population allele frequencies (Fig 4C). A complete summary of population coverage by geographic region is presented in S1 Table. Only alleles scoring in the top 2% of predicted binders were selected for population coverage analysis, which corresponded to ~ 5 alleles/peptide in each group. It is likely that some of the alleles with higher percentile scores (i.e., poorer binding properties) are weak binders under physiological conditions; therefore, we imposed more stringent selection criteria at this stage to limit the overestimation of population coverage that may occur when the full list of predicted binding alleles is evaluated. The candidate peptides were predicted to provide class I coverage for 97.47% of the global population, with 4.09 peptide/HLA combinations recognized on average and PC90 = 2.2, where PC90 is defined as the minimum number of peptide/HLA combinations predicted to be recognized by 90% of the

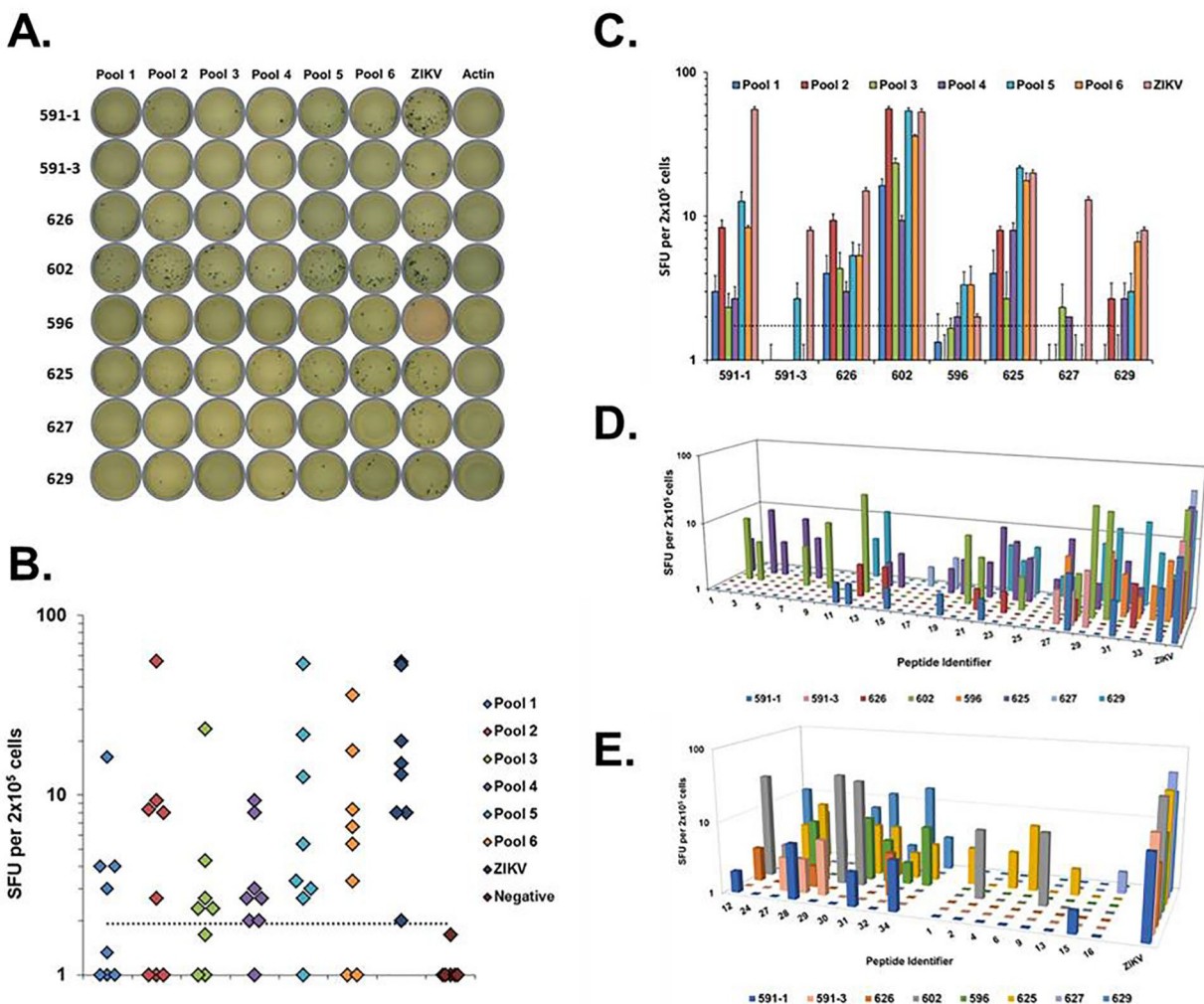

**Fig 3. ZIKV peptide-specific memory T cell responses in convalescent subjects.** (A) Representative images of IFN-γ ELISpot assay plate wells, where each row represents PBMCs from an individual subject and each column represents the indicated stimulus. (B) IFN-γ ELISpot responses to ZIKV peptide pools. Each marker represents the average IFN-γ ELISpot response to the indicated treatment for an individual subject in SFUs/2 x 10⁵ cells. Dotted line marks the cutoff for positive signal detection (2 standard deviations above background). (C) Delineation of peptide pool IFN-γ ELISpot responses by subject. Each bar represents the average IFN-γ ELISpot response of the indicated individual subject against the color-matched peptide pool in SFUs/2x105 cells. Error bars represent the standard error in the measurement. Dotted line marks the cutoff for positive signal detection (2 standard deviations over background). (D) Individual peptide IFN-γ ELISpot responses delineated by subject. Each bar represents the average IFN-γ ELISpot response of an individual subject against the indicated peptide in SFUs/2x105 cells. Chart floor marks the cutoff for positive signal detection (2 standard deviations above background). (E) Average IFN-γ ELISpot response to the candidate peptides (left cluster) and comparator peptides (right cluster). ZIKV responses for all subjects shown for comparison. ZIKV = Zika virus; Negative = actin.

population. Notably, the candidate peptides covered 97.75% of the North American population (4.08 hit average, PC90 = 2.24) and 99.39% of the European population (4.73 hit average, PC90 = 2.81), which are two major population groups that represent geographic areas at risk for the spread of ZIKV. In contrast, the comparator peptide group only covered 89.45% of the global class I alleles, with 3.34 average epitope hits recognized and PC90 = 0.95. Coverage remained high, but coverage decreased for the North American population to 91.1% (3.23 hit average, PC90 = 1.06) and 94.0% (4.05 hit average, PC90 = 1.28) for the European population. Although the number of class II epitopes was limited in both groups, the candidate peptides again displayed more robust population coverage than the comparator group. Global class II

**Table 2. ZIKV candidate and comparator peptide subsets with HLA class I and class II alleles predicted to bind the indicated peptides.**

| Candidate Sequences | | |
|---|---|---|
| **Peptide Sequences** | **Predicted Binding Alleles**[a] | |
| **12/YLQDGLIASL** | Class I | A*02:01, A*02:03, A*02:06, B*15:01, B*08:01, A*24:02, A*23:01, B*51:01 |
| | Class II | N/A[#] |
| **24/ TVVVGSVKNPMWRGPQRLPVPVN** | Class I | A*02:06, A*02:03, A*68:01, A*11:01, A*03:01, A*30:01, A*68:02, A*02:01, B*15:01, A*31:01, B*07:02, B*08:01, A*33:01, B*53:01 |
| | Class II | DRB1*03:01, DRB1*08:02 |
| **27/FLRFTAIKPSLG** | Class I | B*51:01, B*08:01, A*68:01, A*11:01, A*30:01, A*03:01, A*32:01, B*35:01, A*31:01, B*08:01, A*02:03, A*68:02, B*53:01, A*01:01, B*58:01, B*57:01, A*02:03, A*26:01 |
| | Class II | DRB1*01:01, DRB1*07:01, DRB1*04:05, DRB1*09:01, DRB1*15:01, DRB1*11:01, DRB1*08:02 |
| **28/QVASAGITY** | Class I | B*15:01, B*35:01, B*24:02, A*26:01, A*30:02, A*01:01, A*11:01, A*03:01, A*68:01, A*02:06 |
| | Class II | N/A[#] |
| **29/SLINGVVRL** | Class I | A*02:03, A*02:01, A*32:01, A*31:01, A*68:01, B*58:01, A*11:01, B*15:01, B*57:01, A*02:03, A*26:01, A*02:06 |
| | Class II | N/A[#] |
| **30/GGLKRLPAGLLLGHGPI** | Class I | A*03:01, B*07:02, A*02:03, A*32:01, B*53:01, A*02:01, A*02:03, A*31:01, B*15:01, B*35:01, B*08:01, B*51:01 |
| | Class II | DRB1*01:01, DRB1*09:01, DRB1*07:01 |
| **31/VVDGDTLK** | Class I | A*68:01, A*11:01, A*03:01, A*26:01 |
| | Class II | N/A[#] |
| **32/DGIEESDLIIPKSLAGP** | Class I | B*51:01, A*01:01, B*40:01, B*44:02, B*44:03, A*11:01, B*07:02, B*58:01, B*08:01, B*53:01 |
| | Class II | None Identified |
| **34/RIIGDEEKY** | Class I | A*30:02, A*26:01 |
| | Class II | N/A[#] |
| Comparator Sequences | | |
| **Peptide Sequences** | **Predicted Alleles** | |
| **1/GRGPQRLPVP** | Class I | B*07:02, B*51:01, A*30:01 |
| | Class II | N/A[#] |
| **2/ALALAIIKY** | Class I | A*11:01, A*03:01, A*30:01, A*30:02, B*15:01, A*26:01, A*31:01, A*02:01, B*44:02, A*68:01, A*01:01 |
| | Class II | N/A[#] |
| **4/RQDQRGSGQVVTY** | Class I | A*30:02, A*32:01, B*58:01, B*15:01, A*30:02, A*32:01, B*57:01 |
| | Class II | None Identified |
| **6/TVTRNAGLVKRR** | Class I | A*68:02, A*26:01, A*11:01, A*03:01, A*68:01, A*24:02, A*02:06, A*02:03, B*08:01, A*23:01, A*01:01 |
| | Class II | DRB1*07:01, DRB1*08:02, DRB1*09:01, DRB1*11:01, DRB1*13:02 |
| **9/YLIPGLQAA** | Class I | A*02:03, A*02:06, A*02:01, B*15:01, A*23:01, A*26:01, A*24:02 |
| | Class II | N/A[#] |
| **13/ LTVVVGSVKNPmGRGPQRLPVPVN** | Class I | A*02:06, A*02:03, A*68:01, A*11:01, A*03:01, A*30:01, A*68:02, A*02:01, B*15:01, A*31:01, B*07:02, B*08:01, A*33:01, B*53:01 |
| | Class II | DRB1*03:01, DRB1*08:02 |

(*Continued*)

**Table 2.** (Continued)

| 15/IIPKSLAGPLSHHNTREGYRTQ | Class I | B*15:01, B*08:01, B*07:02, B*58:01, B*51:01, A*03:01, B*57:01, A*32:01, B*53:01, A*11:01, A*01:01, A*30:02, A*31:01, A*33:01, A*68:01 |
| --- | --- | --- |
| | Class II | None Identified |
| 16/RGPQRLPVPVN | Class I | B*07:02, B*08:01, A*33:01, B*53:01 |
| | Class II | N/A[#] |

[a]Alleles are listed in order of their predicted binding rank from lowest to highest percentile score. Alleles may have been predicted to bind multiple peptides containing the same consensus sequence or a sequence nested within the larger peptide identified by nLC-MS/MS. For simplicity, only the sequence of the peptide identified by nLC-MS/MS is shown.

[#]Indicates peptide sequence below the minimum length required for prediction algorithm.

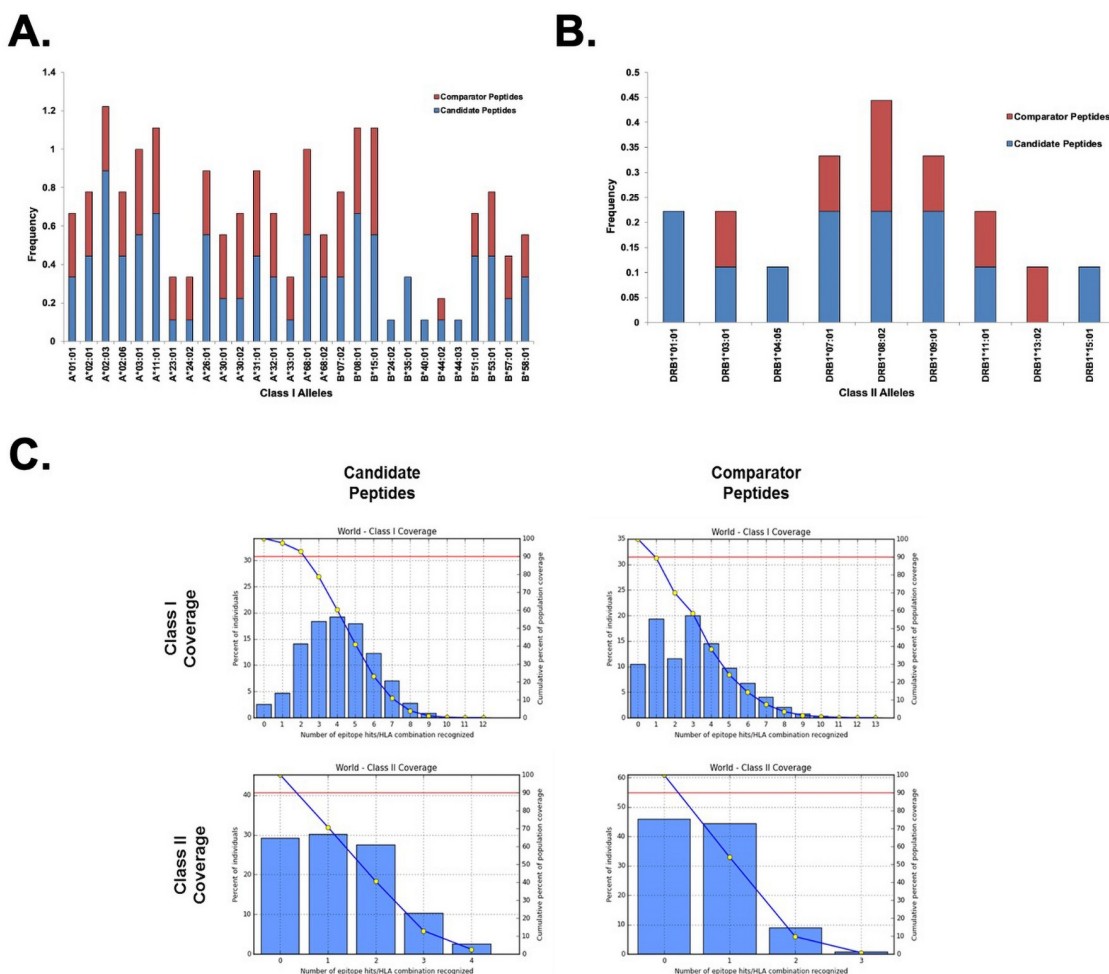

**Fig 4. Predicted allele coverage of select ZIKV peptides.** Coverage of HLA class I (A) and class II (B) alleles of candidate and comparator ZIKV peptides. Frequency represents the fraction of each peptide subset predicted to bind the indicated HLA molecule. (C) Histograms illustrate the fraction of population coverage as a function of the number of HLA/peptide combinations recognized. Blue line trace indicates the cumulative population coverage frequency. Red line indicates the 90th percentile of the global population. Intercept between the two lines represents the minimum number of peptide/HLA combinations recognized by 90% of the global population (PC90).

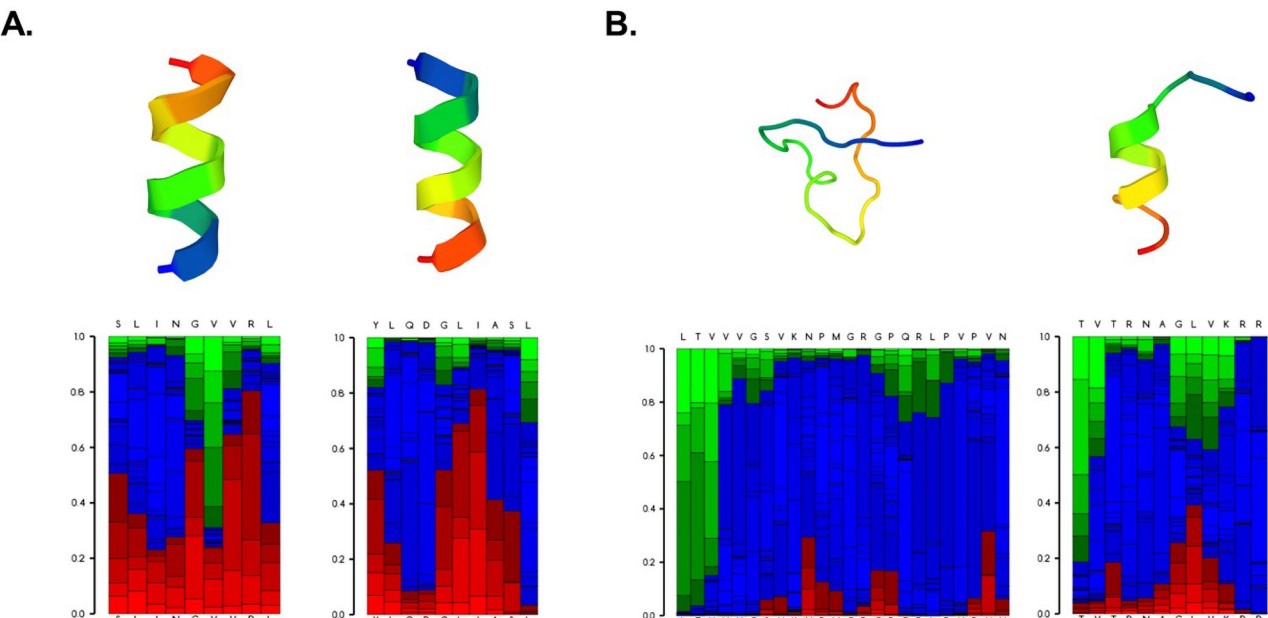

**Fig 5. Predicted structural properties of select ZIKV peptides.** Representative structure predictions for two peptides (29 and 12) in the candidate subset (A) and two peptides (13 and 6) in the comparator subset. Best fit models of the peptide structures are shown in the upper panels. Lower panels indicate the probabilities of local structural properties by amino acid position. Red = helical, blue = coiled, green = extended.

coverage for the candidate peptides was 70.74% (1.27 hit average, PC90 = 0.34) but fell to 54.0% (0.64 hit average, PC90 = 0.22) in the comparator group (S1 Table).

## Modeling of peptide structure and properties

Peptides in the candidate group were largely predicted to have ordered structures that adopted some degree of helical conformation when modeled under physiological conditions, with the exception of peptide 24 (Fig 5, S2 Fig). In contrast, the majority of the comparator peptides (five out of eight) adopted largely disordered structures exhibiting high degrees of coiled and extended structural features under the same modeling parameters. The predicted physical properties of the peptides in each subset are summarized in S2 Table. The candidate peptides were predicted to have a longer theoretical half-life than the comparator group (16.21 hours vs. 8.99 hours; $p$ = 0.5) [50], which correlated with the average instability index for the two groups (24.72 vs. 43.75; $p$ = 0.29). A larger instability index ($>$ 40) is indicative of an unstable peptide structure [51]. The candidate peptides were also predicted to have a larger average aliphatic index than the comparator group (125.41 vs. 103.67; $p$ = 0.34), which is a positive metric for thermostability [52]. Interestingly, the grand average of hydropathy (GRAVY) index differed between the candidate and comparator groups (0.27 vs.– 0.24; $p$ = 0.3), indicating that the comparator peptides were slightly more hydrophilic. Peptide stability is a critical aspect to consider in the design of peptide-based vaccines and therapeutics, and collectively, these modeling data suggest the candidate peptides would be largely stable for formulation and testing as components of a candidate ZIKV vaccine.

## Discussion

Zika virus remains a concern for global public health due to the potential impact of neurological and fetal malformations that could develop should the disease spread and become

established in regions with immunologically naïve populations (e.g., North America, Europe). In an effort to develop safe and widely effective peptide-based ZIKV vaccine components, our primary goal in this work was an exploratory analysis of potential vaccine targets and not to definitively characterize the HLA presentation of the ZIKV peptides identified by our mass-spectrometry approach. Such endeavors will be part of future work where we validate the targets identified here (as well as potential new targets) by using immunoaffinity purification to isolate peptide:HLA complexes. We identified 90 peptides using nLC-MS/MS and rationally narrowed our selection to nine unique peptides that elicited T cell recall immune responses in the majority of convalescent subjects tested. These peptides were subsequently predicted by bioinformatics to provide broad global population coverage and possess stable structural properties for vaccine formulation.

The majority of peptides identified using our *in vitro* ZIKV infection model were derived from non-structural proteins, illustrating the importance of considering both structural and non-structural viral proteins as potential targets for vaccine design [53]. Interestingly, 59 out of 90 peptides identified were derived from the NS1 protein. Previous studies have shown that NS1-based ZIKV vaccines elicit protective cellular and humoral immune responses in mice [54, 55], and mice administered human antibodies against NS1 were protected from ZIKV challenge [56]. In combination with our data, this suggests that NS1 may be a primary target of the cellular immune response to ZIKV, but further studies are warranted to confirm these observations and determine whether anti-NS1 responses mediate protection from disease in humans.

The magnitude and breadth of memory T cell recall responses against ZIKV peptides varied significantly among convalescent subjects, with some subjects exhibiting strong IFN-γ secretion in response to numerous viral peptides, while other subjects displayed more subdued responses (Fig 3). Nevertheless, our approach allowed us to identify a subset of candidate peptides that were "broadly reactive" (i.e., stimulating recall responses in four out of seven subjects) among our limited cohort (Fig 3E). We compared these candidate peptides with a subset of comparator peptides that stimulated more limited T cell responses. Notably, only two subjects (602 and 625) exhibited strong responses to peptides in the comparator subset (Fig 3E). These two subjects collectively exhibited the strongest memory T cell responses against ZIKV peptides overall, suggesting they may harbor some inherent biological factor (e.g., a specific HLA haplotype) that promotes stronger cellular immune responses to ZIKV. Unfortunately, we lacked sufficient biospecimens to match subject HLA types with peptide-specific T cell responses in our study.

The strength of the T cell response was not dependent on the time since infection among our subjects (S1 Fig), although our ability to detect such effects was hampered due to our limited sample size. However, it should be noted that the magnitude of the T cell response declined significantly between blood draws (~ 100 days) for the lone subject (591) recruited for longitudinal study. A recent study reported declines of humoral immunity (i.e., IgG and neutralizing antibody titers) among convalescent adults from French Polynesia [57]. Our observations suggest that cellular immunity to ZIKV may also be short-lived in some individuals, although a controlled study designed to assess waning cellular immunity against ZIKV in a larger population is warranted.

Due to the limited availability of convalescent biological samples, we employed bioinformatics to further analyze the candidate and comparator peptide subsets, hypothesizing that collective differences in HLA binding or peptide physical properties may explain the differing T cell responses. The candidate peptides were predicted to primarily adopt more organized structures (Fig 5, S2 Fig) and exhibit properties consistent with greater stability compared to the comparator peptides (S2 Table). Increased stability under physiological

conditions would presumably increase the bioavailability of the candidate peptides, therefore promoting increased uptake and activation of immune cells following immunization [58, 59].

The peptides in the candidate pool were also predicted to bind more broadly across HLA class I and class II alleles relative to the comparator peptides (Fig 4, Table 2). Consequently, the candidate peptide subset was predicted to have higher global population coverage (Fig 4C, S1 Table), suggesting that a vaccine formulation containing the candidate peptides would elicit immune responses among a greater proportion of the population relative to the comparator peptides. It should be noted that the global population coverage for class I alleles was high for both subsets (97.5% vs. 89.5%); however, the comparator subset was predicted to provide substantially lower coverage for class II alleles (70.7% vs. 54.0%). The discrepancy in class II allele coverage was due primarily to the broad reactivity predicted for peptide 27, which significantly increased the class II allele coverage of the candidate peptide subset (Table 2). This suggests that a vaccine formulated with the candidate peptides–particularly peptide 27 –may be significantly more efficient at stimulating CD4$^+$ T cells, therefore contributing to both humoral and cellular immune responses against ZIKV.

Our study possessed several key strengths but was not without limitations. The use of an *in vitro* infection model to identify naturally-processed peptides presented on HLA molecules of human antigen-presenting cells reduced selection bias in our initial experiments and allowed us to comprehensively study the peptides presented to the immune system during a natural infection. However, the peptides identified in our study likely do not represent the complete ZIKV immunopeptidome, as convalescent subjects with weak peptide-specific responses in our study still exhibited robust responses to the live virus. Furthermore, different antigen-presenting cell lines (e.g., monocytes, dendritic cells) could be used for peptide selection experiments and may present different peptide repertoires. Our use of PBMCs from convalescent subjects was also a strength of our study, as it allowed us to directly validate the immunological relevance of our identified peptides. However, our samples size was limited by the availability of convalescent biospecimens, and as we were not directly involved with subject recruitment, our access to detailed demographic (e.g., age, sex) and clinical (e.g., disease severity) data was limited. Furthermore, while the reliability of predictive algorithms for HLA class I and class II peptide binding has significantly improved in recent years due to advancements in computational methodologies [60, 61], our *in silico* analyses remain inherently limited without biological validation.

In summary, we used nLC-MS/MS coupled with bioinformatics to identify nine naturally-processed ZIKV peptides for further study as components of an experimental ZIKV vaccine [62]. To our knowledge, this is the first study to directly identify ZIKV peptides processed by human immune cells and validate their immunogenicity in convalescent subjects. While these candidate peptides stimulated memory recall responses in convalescent T cells, their efficacy as vaccine immunogens remains unclear, and animal studies are currently ongoing to address this question. Mixtures of candidate peptides alone may only stimulate cellular immunity, but formulation of these viral peptides with other ZIKV vaccine platforms that elicit humoral immune responses (e.g., recombinant subunits, virus-like particles, nanoparticles) represents a promising approach for developing a safe and effective ZIKV vaccine [53]. Our results represent an initial step towards characterizing the immunopeptidome for ZIKV and illustrate the advantages of including peptide-based components in experimental ZIKV vaccine formulations–perhaps particularly to bolster cellular immune responses. It is critical that ZIKV vaccine development and testing continue so that a safe and effective vaccine is available in advance of the next inevitable disease outbreak.

## Supporting information

**S1 Fig. Correlation of recall T cell responses against live ZIKV with time elapsed since infection.**
(JPG)

**S2 Fig. Predicted structural properties of candidate and comparator ZIKV peptides.** Best fit models for all peptides in the candidate and comparator subsets are shown alongside the probabilities of local structural properties by amino acid position. Red = helical, blue = coiled, green = extended.
(PNG)

**S1 Table. Predicted population coverage for the candidate and comparator peptide subsets across individual population groups as defined by HLA allele frequencies.**
(JPG)

**S2 Table. Calculated physical properties for the candidate and comparator peptide subsets.** Theoretical pI represents the calculated isoelectric point for the peptide. The half-life was calculated based on models of cellular processes in cultured mammalian reticulocytes. The aliphatic index is the relative volume occupied by aliphatic amino acid side chains. GRAVY is the sum of hydropathy values for each amino acid in the peptide sequence.
(PNG)

## Acknowledgments

The authors would like to thank Caroline L. Vitse for editorial assistance with this manuscript; Dr. Daniel J. McCormick and Kenneth L. Johnson for mass spectrometry support and analysis; and Drs. Grace Chen and Julie Ledgerwood for generously providing PBMCs from convalescent subjects.

## Author Contributions

**Conceptualization:** Stephen N. Crooke, Inna G. Ovsyannikova, Richard B. Kennedy, Gregory A. Poland.

**Data curation:** Stephen N. Crooke.

**Formal analysis:** Stephen N. Crooke.

**Funding acquisition:** Stephen N. Crooke, Gregory A. Poland.

**Investigation:** Stephen N. Crooke, Inna G. Ovsyannikova, Richard B. Kennedy, Gregory A. Poland.

**Methodology:** Richard B. Kennedy, Gregory A. Poland.

**Project administration:** Stephen N. Crooke, Inna G. Ovsyannikova, Gregory A. Poland.

**Resources:** Gregory A. Poland.

**Supervision:** Inna G. Ovsyannikova, Richard B. Kennedy, Gregory A. Poland.

**Writing – original draft:** Stephen N. Crooke.

**Writing – review & editing:** Stephen N. Crooke, Inna G. Ovsyannikova, Richard B. Kennedy, Gregory A. Poland.

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
