## [Decision Letter · Decision Letter 0]

25 Feb 2021

PONE-D-21-03126

Identification of naturally processed Zika virus peptides by mass spectrometry and validation of memory T cell recall responses in Zika convalescent subjects

PLOS ONE

Dear Dr. Poland,

Thank you for submitting your manuscript to PLOS ONE. After careful consideration, we feel that it has merit but does not fully meet PLOS ONE’s publication criteria as it currently stands. Therefore, we invite you to submit a revised version of the manuscript that addresses the points raised during the review process.

We look forward to receiving your revised manuscript.

Kind regards,

Scheherazade Sadegh-Nasseri

Academic Editor

PLOS ONE

Journal Requirements:

2. Please state where you obtained the ZIKV (Puerto Rico strain PRVABC59) viral sample used in this study. Specifically please state whether the sample was:

(1) from a commercial source (if so please provide the source and catalog number)

(2) from an established biobank (if so please provide the name and a link)

(3) specifically collected for this study or not

(4) collected through a medically prescribed test

(5) completely de-identified before researchers accessed the samples

3. In your Methods section, please provide additional details regarding the B cell line (Priess) used in your study. Please include the source from which you obtained the cells, the catalog number if applicable, whether the cell line was verified, and if so, how it was verified. For more information on PLOS ONE's guidelines for research using cell lines, see https://journals.plos.org/plosone/s/submission-guidelines#loc-cell-lines.

4.  Thank you for stating the following in the Financial Disclosure section:

"The research presented here was supported by funding from the Mayo Clinic Department of General Internal Medicine and the Maurice R. Hilleman Early-Stage Career Investigator Award to SNC (National Foundation for Infectious Diseases and Merck & Co. Inc.).  The funders had no role in study design, data collection and analysis, decision to publish, or preparation of the manuscript. "

We note that you received funding from a commercial source:Merck & Co. Inc.

"Dr. Poland is the chair of a Safety Evaluation Committee for novel investigational vaccine trials being conducted by Merck Research Laboratories. Dr. Poland offers consultative advice on vaccine development to Merck & Co., Medicago, GlaxoSmithKline, Sanofi Pasteur, Emergent Biosolutions, Dynavax, Genentech, Eli Lilly and Company, Johnson & Johnson/Janssen Global Services LLC, Kentucky Bioprocessing, AstraZeneca, and Genevant Sciences, Inc. Drs. Poland and Ovsyannikova hold patents related to vaccinia and measles peptide vaccines. Dr. Kennedy holds a patent related to vaccinia peptide vaccines. Drs. Poland, Kennedy, and Ovsyannikova have received grant funding from ICW Ventures for preclinical studies on a peptide-based COVID-19 vaccine.  Dr. Kennedy has received funding from Merck Research Laboratories to study waning immunity to mumps vaccine. These activities have been reviewed by the Mayo Clinic Conflict of Interest Review Board and are conducted in compliance with Mayo Clinic Conflict of Interest policies. This research has been reviewed by the Mayo Clinic Conflict of Interest Review Board and was conducted in compliance with Mayo Clinic Conflict of Interest policies. All other authors declare no competing financial interests."

7. We note that you have included the phrase “data not shown” in your manuscript. Unfortunately, this does not meet our data sharing requirements. PLOS does not permit references to inaccessible data. We require that authors provide all relevant data within the paper, Supporting Information files, or in an acceptable, public repository. Please add a citation to support this phrase or upload the data that corresponds with these findings to a stable repository (such as Figshare or Dryad) and provide and URLs, DOIs, or accession numbers that may be used to access these data. Or, if the data are not a core part of the research being presented in your study, we ask that you remove the phrase that refers to these data.

8. Please upload a copy of Supporting Information Figure 2 and Supporting Information Table 2 which you refer to in your text (lines 476 and 481).

Reviewers' comments:

Reviewer's Responses to Questions

**Comments to the Author**

1. Is the manuscript technically sound, and do the data support the conclusions?

Reviewer #1: Partly

Reviewer #2: Partly

2. Has the statistical analysis been performed appropriately and rigorously? 

Reviewer #1: Yes

Reviewer #2: N/A

3. Have the authors made all data underlying the findings in their manuscript fully available?

Reviewer #1: Yes

Reviewer #2: No

4. Is the manuscript presented in an intelligible fashion and written in standard English?

Reviewer #1: Yes

Reviewer #2: Yes

5. Review Comments to the Author

Reviewer #1: The authors have produced an interesting work trying to identify processed peptides that could activate a cellular immune response against Zika virus. The overall conclusions are significant, and I think contribute to the overall topic of generating a potential vaccine against this disease.

I feel that the following areas require a response and defense from the authors:

1- Identified peptides using a cell lines that are homozygous for HLA-A*02:01 and the HLA-B that this cell expressed (data do not show in the manuscript) only generate peptides presented by these molecules and for instance only produce a proper T-cell response in those individuals that expressed the same alleles of MHC-I molecules

2- Although those peptides could be presented by another MHC-I molecules as you explain in the manuscript, what is the affinity of those peptides to other MHC-I molecules?

3- Could you explain how those peptides generated for some MHC-I molecules could immunize people that do not present the MHC-I molecules that could bind those potential peptides?

4- Why do you not genotype the patients whose plasma is used in the T-cell ELISPOT in order to explain how some patients have a strong immune response against the peptides while others do not?

5- Do you not think that based only on in silico preditions could have limitations in order to clarify the immune response against those peptides?

Reviewer #2: The manuscript by Crooke et al. shows the identification by mass spectrometry of HLA ligands derived from Zika virus and the study of memory T cell response in Zika convalescent subjects. The study of T cell epitopes from Zika virus, and other viral pathogens, is of outstanding interest at this moment and are required for the design of future vaccines.

However, I have some concerns regarding some points of the manuscript and the how the study was made:

MAJOR CONCERNS:

1. Acid stripping treatment works very well for HLA-I molecules, but it is not as good for HLA-II molecules (as they are stabilized at low pH). Authors should show that with this protocol, ligands from HLA-II molecules are obtained.

1. The putative HLA ligands was purified after the treatment of Priess cells, after two washes with PBS, with an acid buffer, with the consequent peptide stripping. The fact that no HLA immunoprecipitation was done and as mass spectrometry is a highly sensitive technique, any viral peptide remaining in the medium would probably be detected. The authors indicate that a total of 2305 MS/MS spectra were analyzed. However, they do not define the criteria to consider a peptide as correct (Mascot score, etc). In addition, the authors do not show the number of peptides (from human, bovine and Zika virus) with a correct assigned sequence. Thus, it is important that authors add a supplementary table with the list containing the sequence of all peptides identified, the parameters of the search (Mascot score, etc) and the assignation as a binder of one HLA molecule expressed by the cells. This is important to see if the peptides contain the anchor motifs to bind to the HLA molecules expressed in Priess cells (HLA-A*02:01, -B*15 and DRB1*04:01).

3. The HLA binding prediction was made with NetMHCpan 4.0 for HLA-I and NetMHCIIpan 3.2 for HLA-II. This is software widely used for this goal. However, the authors say that "The complet list of predicted peptides was filtered for sequences matching (or nested within) selected peptide sequences..." both for HLA-I and HLA-II. I consider that it is correct for HLA-II, but not for HLA-I, as the peptide ends are fixed and interacting with the HLA molecule. I would like to know the explanation to this sentence as maybe I have not understood how the filter was applied.

4. I do not understand the Table 2. For example, peptide 24 is clearly too long to bind with enough affinity to any HLA class I molecule. Thus, I consider that this peptide should not be considered as a putative binder for HLA-I molecules. The predicted binding affinity should be shown (the "Affinity (nM)" value is a good parameter to indicate the affinity of a peptide for a specific HLA molecule).

5. The manuscript lacks supplementary figure 2 and table 2. At least, I could not see them.

MINOR CONCERNS:

1. The complete HLA typing of Priess cells should be shown in the manuscript, as other HLA molecules than A*02:01 or DRB1*04:01 can present peptides on the cell surface.

6. PLOS authors have the option to publish the peer review history of their article (what does this mean?). If published, this will include your full peer review and any attached files.

Reviewer #1: **Yes: **Adrián Martin-Esteban

Reviewer #2: No

---

## [Author Response · Author response to Decision Letter 0]

30 Mar 2021

Dear Scheherazade Sadegh-Nasseri:

Thank you for these helpful comments from reviewers. We have considered them in depth and responded to each comment or request in detail. We trust these responses will prove acceptable and look forward to your response.

The point-by-point response to reviewers’ comments can be found below.

Reviewers’ comments:

Reviewer #1: 

Comments to the authors:

The authors have produced an interesting work trying to identify processed peptides that could activate a cellular immune response against Zika virus. The overall conclusions are significant, and I think contribute to the overall topic of generating a potential vaccine against this disease.

I feel that the following areas require a response and defense from the authors:

1. Identified peptides using a cell lines that are homozygous for HLA-A*02:01 and the HLA-B that this cell expressed (data do not show in the manuscript) only generate peptides presented by these molecules and for instance only produce a proper T-cell response in those individuals that expressed the same alleles of MHC-I molecules

Authors’ response: We direct the reviewer to our revised statements in the Methods section regarding the other HLA alleles for the Priess cell line. For the HLA-B in question, Priess cells are documented to express HLA-B*15. While we would certainly predict a strong T cell response in individuals sharing the same alleles as the Priess cell line, some of the peptides identified by our experimental approach are likely to bind and be presented by more than one HLA molecule (e.g., other alleles within an HLA supertype)—thus stimulating responses in a range of individuals and not just those with HLA types that are matched to Priess cells. 

2. Although those peptides could be presented by another MHC-I molecules as you explain in the manuscript, what is the affinity of those peptides to other MHC-I molecules

Authors’ response: We refer the reviewer to Table 2 in our manuscript for the detailed list of HLA class I molecules that peptides are predicted to bind based on our approach. While the predicted affinities range widely for each peptide-HLA interaction, most peptides were predicted to bind with low nanomolar to low micromolar affinities, which is reflected by the percentile cutoffs used in our Methods section. We note that we selected our predicted peptides with a conservative percentile rank threshold, which selected for the best binders (e.g., those with the highest affinity). However, we should also note that predictive algorithms are not a perfect representation of actual biology, and the true binding affinities of the peptides listed here may differ in vivo. 

3. Could you explain how those peptides generated for some MHC-I molecules could immunize people that do not present the MHC-I molecules that could bind those potential peptides?

Authors’ response: We disagree that peptides presented by one HLA allele cannot bind to and be presented by other HLA alleles. Peptide promiscuity in HLA presentation clearly happens and is the entire premise behind HLA supertypes. Please note that while the peptides identified by mass spectrometry in our study would have only been isolated from the HLA alleles expressed by the Priess cells, our immunoinformatic analysis identified numerous other HLA alleles that these peptides are predicted to bind with relatively high affinities. We used these predicted alleles for our calculations of population coverage based on known allele frequencies. From this, we determined that the candidate peptides listed in Table 2 provided >97% global coverage for HLA class I alleles and >70% global coverage for HLA class II alleles. This is supportive evidence that a vaccine formulated with these peptides could be used to immunize a significant proportion of the population against Zika. While such a vaccine may not be as effective in certain individuals (based on their HLA type), the threshold for herd immunity could easily be reached if the vaccine achieved this level of coverage in the population. Further experimentation in vitro and in vivo are obviously warranted to confirm these predictions. 

4. Why do you not genotype the patients whose plasma is used in the T-cell ELISPOT in order to explain how some patients have a strong immune response against the peptides while others do not?

Authors’ response: This is an excellent suggestion. We sought to do this but were provided with a very limited quantity of PBMCs for these subjects. We exhausted our supply of PBMCs conducting and verifying the results of the ELISpot experiments, so we did not have any biospecimens remaining to conduct genotyping (see Lines 362-363). This is something that we would like to accomplish in future work to more directly relate the immunological response with a subject’s HLA type.

5. Do you not think that based only on in silico preditions could have limitations in order to clarify the immune response against those peptides?

Authors’ response: We certainly agree that in silico predictions are not without limitations; however, given our limited accessibility to biospecimens from ZIKV convalescent subjects, we felt that an immunoinformatics analysis was best suited to address whether the peptides identified by our mass spectrometry-based approach may have the potential to elicit a response in individuals with different HLA types. By combining predictive information on peptide binding with estimates of population coverage for different HLA alleles, we feel that the results of our study provide preliminary evidence of peptides that may be of great interest in vaccine development studies due to their putatively broad population reactivity. 

Reviewer #2

Comments to the authors:

The manuscript by Crooke et al. shows the identification by mass spectrometry of HLA ligands derived from Zika virus and the study of memory T cell response in Zika convalescent subjects. The study of T cell epitopes from Zika virus, and other viral pathogens, is of outstanding interest at this moment and are required for the design of future vaccines.

However, I have some concerns regarding some points of the manuscript and the how the study was made:

MAJOR CONCERNS:

1. Acid stripping treatment works very well for HLA-I molecules, but it is not as good for HLA-II molecules (as they are stabilized at low pH). Authors should show that with this protocol, ligands from HLA-II molecules are obtained.

Authors’ response: A method for isolation of immunogenic peptides from HLA class I complexes was originally proposed by Storkus in 1993 (see Storkus WJ et al, J of Immunotherapy, 1993). While we acknowledge that acid stripping is more efficient for isolating peptides associated with HLA-I molecules (as the reviewer points out), this approach has been previously shown to be an effective method for stripping peptides from HLA-II molecules (see Ramachandra et al, J Immunol 1999). We emphasize that our primary goal in this work was an exploratory analysis of potential vaccine targets and not to definitively characterize the HLA presentation of the ZIKV peptides identified by our mass-spectrometry approach. Such endeavors will be part of future work where we validate the targets identified here (as well as potential new targets) by using immunoaffinity purification to isolate peptide:HLA complexes. 

2. The putative HLA ligands was purified after the treatment of Priess cells, after two washes with PBS, with an acid buffer, with the consequent peptide stripping. The fact that no HLA immunoprecipitation was done and as mass spectrometry is a highly sensitive technique, any viral peptide remaining in the medium would probably be detected. The authors indicate that a total of 2305 MS/MS spectra were analyzed. However, they do not define the criteria to consider a peptide as correct (Mascot score, etc). In addition, the authors do not show the number of peptides (from human, bovine and Zika virus) with a correct assigned sequence. Thus, it is important that authors add a supplementary table with the list containing the sequence of all peptides identified, the parameters of the search (Mascot score, etc) and the assignation as a binder of one HLA molecule expressed by the cells. This is important to see if the peptides contain the anchor motifs to bind to the HLA molecules expressed in Priess cells (HLA-A*02:01, -B*15 and DRB1*04:01).

Authors’ response: We believe the level of detail provided in the “MS/MS data analysis” subsection of the Methods sufficiently outlines our workflow for identifying peptides from ZIKV (as well as peptides of human and bovine origin) and correctly assigning their origin. We elected not to include the full list of peptide sequences for two reasons: 1) the list files were incredibly large, and we felt their inclusion did not significantly contribute to the overall findings of our report; and 2) several of the ZIKV peptides not presented in this work are components of active research that will be publicly disclosed in a later publication. We again stress that our primary goal was to identify preliminary targets for a peptide-based vaccine, and we should note that any given peptide sequence may bind to more than one HLA molecule (as noted in the predictions from our immunoinformatic analysis). While we concede that the reviewer is correct in noting that remnant viral peptides in the media that were not HLA associated may have been identified, we note two things: 1) we used size-exclusion filtration to isolate peptides from remnant HLA molecules and other proteins in the wash buffer—this step would have also presumably removed any remaining source of viral peptides (e.g., viral particles, viral proteins) that were not already digested and processed; and 2) the list of peptides that we carried forward from our mass-spectrometry analysis were those with the highest confidence identified in two biological replicate experiments. We have added a statement in the Methods to make this clearer. Finally, the ability of these peptides to elicit T cell recall responses in infected subjects demonstrates that those peptides we identified as promising candidates are indeed capable of being recognized by HLA molecules. 

3. The HLA binding prediction was made with NetMHCpan 4.0 for HLA-I and NetMHCIIpan 3.2 for HLA-II. This is software widely used for this goal. However, the authors say that "The complet list of predicted peptides was filtered for sequences matching (or nested within) selected peptide sequences..." both for HLA-I and HLA-II. I consider that it is correct for HLA-II, but not for HLA-I, as the peptide ends are fixed and interacting with the HLA molecule. I would like to know the explanation to this sentence as maybe I have not understood how the filter was applied.

Authors’ response: We apologize for the confusion and have amended our statement to be clearer on the purpose of filtering the sequences. The predictions we ran identified peptides across the entire sequence of the ZIKV protein(s). Our intent here was to down-select from the computational predictions to identify consensus sequences that were matched to peptides that we identified by mass-spectrometry analysis. These consensus sequences could have been either an exact match to a peptide we identified or partially/fully contained within a peptide that we identified. Because exact matches between the predicted and mass-spectrometry-identified peptides were not present for all peptides, we used the consensus sequence of the predicted peptide to inform our decision for selecting peptides identified by mass spectrometry that had partial sequence homology. Our approach was such that using peptides identified from in vitro experiments (i.e., the peptides identified by mass spectrometry) was of higher priority than using peptides identified from the binding predictions, as any computational method may not completely replicate true biological processes and therefore may not identify all peptides that can be identified experimentally. 

4. I do not understand the Table 2. For example, peptide 24 is clearly too long to bind with enough affinity to any HLA class I molecule. Thus, I consider that this peptide should not be considered as a putative binder for HLA-I molecules. The predicted binding affinity should be shown (the "Affinity (nM)" value is a good parameter to indicate the affinity of a peptide for a specific HLA molecule).

Authors’ response: We agree with the reviewer that the length of certain peptides identified by our mass-spectrometry analysis would preclude them from binding certain HLA molecules. While the prediction algorithms identified multiple peptides containing the same or similar consensus sequences, we did not identify all variations of these peptides by mass spectrometry, and we certainly did not have the available samples to synthesize and test all predicted sequences. Therefore, the larger sequences shown in Table 2 (presumably HLA-II binders) are peptides that were identified by mass spectrometry, and reactivity with HLA-I alleles was noted as smaller sequences nested within this parent sequence were predicted to bind HLA-I molecules. Because we do not know how these peptides would be processed upon delivery in a vaccine formulation, we opted to err on the side of caution and present HLA alleles from both class I and class II that were predicted to bind any part of these peptides. We have added a footnote to Table 2 for clarification. 

We feel that the scoring threshold applied was a much more comprehensive metric to evaluate peptide binding as it incorporated binding affinity and several other metrics (as detailed on the webpages for each of the algorithms used). For clarification, all peptides discussed for detailed analysis in this report had binding affinities in the low nanomolar to low micromolar range for the various alleles. 

5. The manuscript lacks supplementary figure 2 and table 2. At least, I could not see them.

Authors’ response: We apologize for this omission. Figure S2 and Table S2 are uploaded with the revised version of the paper.

MINOR CONCERNS:

1. The complete HLA typing of Priess cells should be shown in the manuscript, as other HLA molecules than A*02:01 or DRB1*04:01 can present peptides on the cell surface.

Authors’ response: We have added a descriptive statement in the Methods that details the complete HLA typing of the Priess cell line (as documented by Millipore Sigma and the ECACC). 

End of response to reviewers’ comments.

We trust these responses will prove satisfactory upon review and look forward to having the manuscript accepted. We thank the reviewers for their careful and insightful comments. We believe the manuscript is stronger for them.

Sincerely,

Gregory A. Poland, M.D.

---

## [Decision Letter · Decision Letter 1]

5 May 2021

PONE-D-21-03126R1

Identification of naturally processed Zika virus peptides by mass spectrometry and validation of memory T cell recall responses in Zika convalescent subjects

PLOS ONE

Dear Dr. Poland,

Thank you for submitting your manuscript to PLOS ONE. After careful consideration, we feel that it has merit but does not fully meet PLOS ONE’s publication criteria as it currently stands. Therefore, we invite you to submit a revised version of the manuscript that addresses the points raised during the review process.

We look forward to receiving your revised manuscript.

Kind regards,

Paulo Lee Ho, Ph.D.

Academic Editor

PLOS ONE

Journal Requirements:

Reviewers' comments:

Reviewer's Responses to Questions

**Comments to the Author**

1. If the authors have adequately addressed your comments raised in a previous round of review and you feel that this manuscript is now acceptable for publication, you may indicate that here to bypass the “Comments to the Author” section, enter your conflict of interest statement in the “Confidential to Editor” section, and submit your "Accept" recommendation.

Reviewer #1: All comments have been addressed

Reviewer #2: All comments have been addressed

2. Is the manuscript technically sound, and do the data support the conclusions?

Reviewer #1: Yes

Reviewer #2: Yes

3. Has the statistical analysis been performed appropriately and rigorously? 

Reviewer #1: N/A

Reviewer #2: Yes

4. Have the authors made all data underlying the findings in their manuscript fully available?

Reviewer #1: Yes

Reviewer #2: No

5. Is the manuscript presented in an intelligible fashion and written in standard English?

Reviewer #1: Yes

Reviewer #2: Yes

6. Review Comments to the Author

Reviewer #1: Stephen N Crooke and co-authors have addressed all my concerns adequately. I recommended publication.

Reviewer #2: I want to thank the responses to my previous comments. The authors have cleared most of my doubts out.

My unique request to the authors to consider the manuscript acceptable to publication is to add to the manuscript a sentence similar to that included in their comment to my first question: "...our primary goal in this work was an exploratory analysis of potential vaccine targets and not to definitively characterize the HLA presentation of the ZIKV peptides identified by our mass-spectrometry approach. Such endeavors will be part of future work where we validate the targets identified here (as well as potential new targets) by using immunoaffinity purification to isolate peptide:HLA complexes".

I consider this is important as, in my opinion, the authors have not completely demonstrated in this work that the peptides are bona fide HLA ligands.

7. PLOS authors have the option to publish the peer review history of their article (what does this mean?). If published, this will include your full peer review and any attached files.

Reviewer #1: **Yes: **Adrian Martin Esteban

Reviewer #2: No

---

## [Author Response · Author response to Decision Letter 1]

7 May 2021

Reviewers’ comments:

Reviewer #1: 

Comments to the authors:

Stephen N Crooke and co-authors have addressed all my concerns adequately. I recommended publication.

Authors’ response: We thank the reviewer for the critique of our manuscript and are glad to have addressed their concerns.

Reviewer #2

Comments to the authors:

I want to thank the responses to my previous comments. The authors have cleared most of my doubts out.

My unique request to the authors to consider the manuscript acceptable to publication is to add to the manuscript a sentence similar to that included in their comment to my first question: "...our primary goal in this work was an exploratory analysis of potential vaccine targets and not to definitively characterize the HLA presentation of the ZIKV peptides identified by our mass-spectrometry approach. Such endeavors will be part of future work where we validate the targets identified here (as well as potential new targets) by using immunoaffinity purification to isolate peptide:HLA complexes".

I consider this is important as, in my opinion, the authors have not completely demonstrated in this work that the peptides are bona fide HLA ligands.

Authors’ response: We agree that our work does not prove HLA binding and presentation for the peptides identified in the manuscript. Therefore, we have added the text suggested by Reviewer #2 in the Discussion (Line 382) and have removed the following text: “we focused on peptides that were naturally processed and presented on HLA molecules following viral infection.”

End of response to reviewers’ comments.

We trust these responses will prove satisfactory upon review and look forward to having the revised manuscript accepted. We thank the reviewers for their careful and insightful comments. We believe the manuscript is stronger for them.

---

## [Editor Report · Decision Letter 2]

12 May 2021

Identification of naturally processed Zika virus peptides by mass spectrometry and validation of memory T cell recall responses in Zika convalescent subjects

PONE-D-21-03126R2

Dear Dr. Poland,

We’re pleased to inform you that your manuscript has been judged scientifically suitable for publication and will be formally accepted for publication once it meets all outstanding technical requirements.

Kind regards,

Paulo Lee Ho, Ph.D.

Academic Editor

PLOS ONE
---

## [Editor Report · Acceptance letter]

19 May 2021

PONE-D-21-03126R2 

Identification of naturally processed Zika virus peptides by mass spectrometry and validation of memory T cell recall responses in Zika convalescent subjects 

Dear Dr. Poland:

I'm pleased to inform you that your manuscript has been deemed suitable for publication in PLOS ONE. Congratulations! Your manuscript is now with our production department. 

Kind regards, 

on behalf of

Dr. Paulo Lee Ho 

Academic Editor

PLOS ONE